# Negative effects by mineral accretion technique on the heat resilience, growth and recruitment of corals

Ewout Geerten Knoester[1,2]*, Richard Sanders[1,3,4], Daisy Durden[1,3,4], Bulisa O. Masiga[2], Albertinka J. Murk[1], Ronald Osinga[1]

1 Marine Animal Ecology, Wageningen University & Research, Wageningen, The Netherlands,
2 REEFolution Trust, Diani, Kenya, 3 Department of Zoological Sciences, Kenyatta University, Nairobi, Kenya, 4 Department of Earth Sciences, Utrecht University, Utrecht, The Netherlands

* ewout.knoester@wur.nl

**Data Availability Statement:** Data and relevant code for this research work are stored in GitHub: https://github.com/ewoutknoester/MineralAccretionBleaching and have been archived

## Abstract

Restoration and artificial reefs can assist the recovery of degraded reefs but are limited in scalability and climate resilience. The Mineral Accretion Technique (MAT) subjects metal artificial reefs to a low-voltage electrical current, thereby creating a calcium-carbonate coating. It has been suggested that corals on MAT structures experience enhanced health and growth. However, prior studies report conflicting results potentially due to different conditions, species and approaches used. We investigated how MAT influences the bleaching resilience, condition and growth of four coral species and natural coral recruitment in Kenya. Coral fragments were outplanted on charged iron tables using commonly-applied settings (6 V; 0.84 A m$^{-2}$). After one month, when all tables had acquired a calcium-carbonate coating, half of the tables were taken off electricity to serve as controls. Both treatments (MAT and Control) were monitored on coral brightness, condition (live tissue cover), growth and natural recruitment for one year, during which a marine heatwave occurred. Coral bleaching was significantly more severe on MAT for all studied species. For three species, coral condition dropped sharply during the heatwave and this decline was faster and more severe on MAT. Coral growth was reduced during the heatwave for all corals and remained low for one species on MAT. After one year, the Control harboured 34 coral recruits, whereas none were found on MAT. Thus, while MAT can be useful to prevent corrosion of metal artificial reefs, we do not recommend MAT as reported here to improve coral growth, condition, heat resilience or recruitment.

## Introduction

Globally, coral reefs and their associated ecosystem services have declined strongly due to both local stressors and climate change [1]. A combination of urgent climate action and improved local management is advocated to preserve current reefs and to improve prospects for natural recovery and reef rehabilitation [2]. Active reef rehabilitation has only gradually become

within a Zenodo repository: https://doi.org/10.5281/zenodo.13164079.

**Funding:** The author(s) received no specific funding for this work.

**Competing interests:** The authors have no competing interests to declare that are relevant to the content of this article.

accepted by the scientific community as a credible conservation tool and therefore ample opportunities remain to improve this management intervention [3, 4]. A commonly-used rehabilitation method is coral gardening, where coral fragments are grown in nurseries before being outplanted onto degraded or artificial reefs [5, 6]. While coral gardening has been shown to improve coral cover and reef biodiversity locally [7, 8], there remain concerns about the feasibility and costs of upscaling this labour-intensive method [9]. Furthermore, if the imminent rise in seawater temperature exceeds the thermal threshold of outplanted coral colonies, the rehabilitation efforts are likely to be nullified by coral bleaching and mortality [2, 10]. These challenges have spurred research into the development of innovative rehabilitation methods with improved cost-effectiveness and thermal resilience [11].

One such potential method is the Mineral Accretion Technique (MAT), previously patented as Biorock® [12]. The premise of MAT lies in the electrolysis of seawater: a low-voltage electric potential is established between a negative cathode (typically a larger iron artificial reef) and a positive anode (typically a small object made of titanium or another noble metal). The reduction of seawater around the cathode creates an alkaline environment, which facilitates the precipitation of calcium carbonate on the iron structure. It has been hypothesized that coral fragments attached to the cathode can achieve faster skeletal growth, as the alkaline environment might reduce the energetic costs needed for corals to deposit their calcium-carbonate skeleton [12]. In addition, the increased availability of electrons around the cathode have been hypothesized to increase the metabolic efficiency of corals, leaving the corals with more energy available for reproduction or to withstand stressors such as heat [12]. Combined, these putative benefits of faster growth and higher stress tolerance would allow MAT to assist in overcoming common challenges associated with coral gardening by improving the cost-effectiveness and climate resilience of reef rehabilitation techniques.

The evidence in support of enhanced physiological coral performance on MAT, however, remains inconclusive. Anecdotal field observations mention up to five times higher coral growth rates [13], 50 times higher survival during heat stress [14] and 1000 times higher natural coral settlement on MAT structures [15]. Such remarkable improvements could not be replicated in controlled experiments and peer-reviewed studies, which either showed only small or temporary improvements in coral growth [16, 17], mixed positive and negative effects on coral performance [18–22] or even strong negative effects, for example on coral recruitment [23]. The transience of any positive effects of MAT could potentially be explained by the insulating effect of the accreted calcium carbonate layer on the cathode, but long-term (>6 month) MAT studies to confirm this remain scarce. The reported mixed effects could indicate that the effectiveness of MAT is strongly dependent on the specific species, environment or setup. However, for a surprisingly large number of the aforementioned studies, the positive effects on coral performance might also have been inadvertently ascribed to MAT due to poorly designed controls. For example, in several studies the better coral performance on MAT compared to control structures could also be explained by the corrosion of the uncoated iron control structures [e.g. 20, 24, 25] or higher sedimentation and predation on natural coral colonies used as 'controls' [26]. Well-designed, multi-species and long-term studies would help to resolve the uncertainty around the effectiveness of MAT.

The aim of this study was to investigate the effects of MAT on the growth, condition (i.e. live coral tissue), heat resilience and recruitment of corals. Fragments of four different coral species were outplanted in southern Kenya on (pre-coated) iron structures with or without electric current and monitored for one year, during which a natural marine heatwave took place. Considering the hypothesized benefits of the alkaline environment and free availability of electrons in combination with the (anecdotal) reports of positive effects, we expected better coral growth, condition and heat resilience on MAT. These benefits were expected to be

species-specific and to dissipate throughout the one-year study duration. Given the relatively fast precipitation of calcium carbonate on the cathode, we expected lower coral recruitment on MAT due to the envelopment of new recruits by the encroaching coating.

## Methods

### Study setting

This study was performed from December 2019 to December 2020 at Firefly House Reef (-4.6505, 39.3866) in Shimoni, Kenya. Average long-term sea surface temperatures range from 25°C in August to 29°C in April. Typically, salinity varies between 34–35, pH between 7.8–8.2 and conductivity between 39,000–49,000 μS cm$^{-1}$ [27]. The study site was situated in a kilo-metre-wide sea strait that is subjected to semi-diurnal tides, with tidal differences of up to 4 m and an average visibility of around 6 m. The substrate consisted of a mix of coral rubble, sea-grass meadows and coral patches with an average hard coral cover of around 50% [28]. This research was implemented as part of an ongoing community-led coral reef restoration project by the Kenyan NGO REEFolution and the local Beach Management Unit, who have placed numerous coral nurseries and artificial reef structures at the site since 2016. The research was undertaken under the national Kenyan research license NACOSTI/21/8896. Additional infor-mation regarding the ethical, cultural, and scientific considerations specific to inclusivity in global research is included in the S1 Checklist. The help of an additional NGO was sought for the setup of the MAT system. This NGO, which prefers to remain anonymous, has eight years hands-on experience with setting up MAT systems around the world.

### Experimental setup

Two treatments were established: the MAT treatment (Fig 1A), for which coral fragments were attached to nine iron tables that were continuously supplied with a low-voltage current. The Control treatment (Fig 1B) featured an identical setup, but was supplied with electricity only for the first month and was then disconnected and moved outside the electrical field of MAT structures for the remainder of the study. The Control treatment was connected to the electric grid for the first month in order to enable the deposition of a calcium-carbonate layer on the tables, thus preventing any confounding effects of corrosion on coral performance on the Control tables. Both treatments were deployed at 1 m depth (low tide) and connected to the electric grid on 17–19 December 2019 (Fig 1C). Subsequently, the Control treatment was disconnected and moved 35 m away from the MAT setup on 25 January 2020 (see S1 and S2 Figs for a map and in-situ pictures of the setup). To replace the Control tables and prevent overcharging of the remaining MAT tables, an additional nine metal structures (Pod MAT) were connected to the grid. These pod MAT structures are not taken into further consider-ation in this study.

   The iron tables for the MAT and Control treatments were made from 12 mm rebar: a square 85-cm table frame was supported by four 70-cm legs at each corner, and eight 40-cm supporting beams reinforced the legs. In addition, a 4 x 4 grid of 8-mm rebar was placed on top, allowing 16 coral fragments to be outplanted on the intersections. The surface area of one table totalled to 0.54 m$^2$. Power was supplied through a 235 V power socket and transformed to 10 V and 8.2 A using an AC/DC converter. Thus, the current density (i.e. cathodic surface area divided by the amperage) for a single MAT structure was 0.84 A m$^{-2}$. A cylinder (40 cm length, 10 cm diameter) of titanium mesh was used as anode and suspended 1 m above the sea-floor at the centre of all MAT tables, which were placed in a circle (Fig 1A), keeping an equal distance of around 3 m between each table and the anode. MAT tables were separated by at least 1 m from each other. Due to the large tidal range, a 100-m armoured cable (4 core

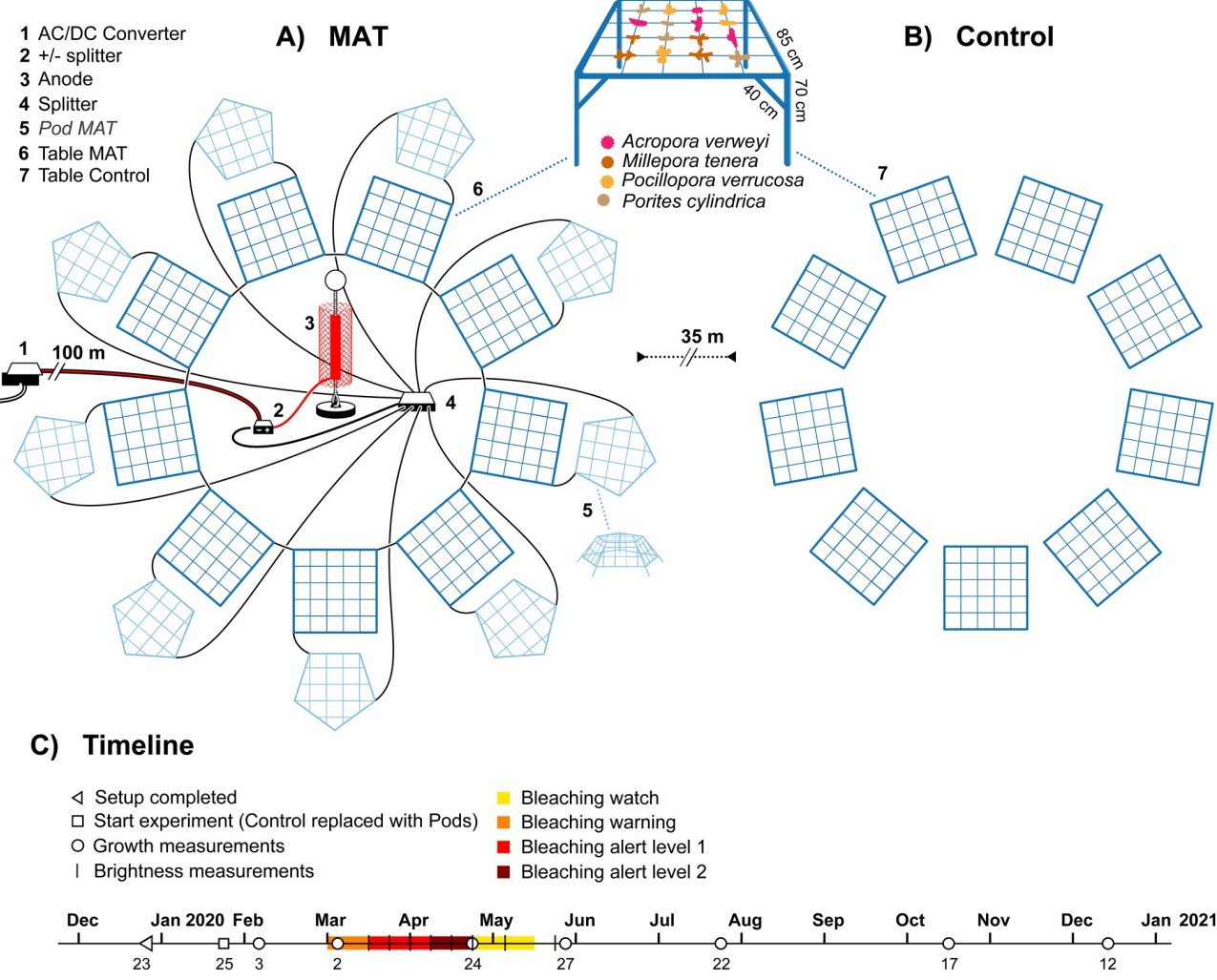

**Fig 1. A schematic diagram of the experimental setup.** A) The mineral accretion technique (MAT) and B) the Control treatment setup and C) a timeline of the study, indicating the measurements and the progression of a marine heatwave (based on NOAA's Coral Reef Watch). During the first month of the study (23 December 2019 to 25 January 2020), both the MAT and Control tables (with the coral fragments attached) were connected to the electric grid to provide an anti-corrosion coating. On 25 January 2020, the Control tables were taken off the grid and moved 35 m away. To keep current densities at the appropriate level after the removal of Control tables, new structures (Pod MAT) were attached to the electrical grid, but these were not further considered during this study. One experimental table is enlarged to show its dimensions and the attachment of the four coral species. Artwork by Vrijlansier.

3 mm$^2$) was needed to bridge the distance between the convertor and the MAT setup. Based on the resistance of the cable it was calculated that around 6.4 V should reach the experimental setup. Similar setups have given the best empirical results in earlier projects and the power values are well within the suggested range by the original inventors, who recommended a current density between 0.1–30 A m$^{-2}$ and voltage between 3–15 V [12]. This setup was further validated by the observed calcium-carbonate accretion rate of 1.3 cm y$^{-1}$ (S3 Fig), which lies neatly within the recommended accretion rate of 1–2 cm y$^{-1}$ [29]. Except during short (<3 h) power outages which occurred a few times a month, the setup remained fully functional throughout study duration, as confirmed through regular (at least monthly) checks. During checks, macroalgae were brushed off the tables to prevent any confounding effects of fouling on coral performance.

Four species with branching growth form were selected for this study and harvested from nearby nurseries: the reef-building hydrozoan *Millepora tenera* and the hard corals *Acropora verweyi*, *Pocillopora verrucosa* and *Porites cylindrica*. These species were locally abundant and represent different life history strategies [30]: both *A. verweyi* and *M. tenera* are highly competitive species that grow quickly, *P. verrucosa* is a generalist species and *P. cylindrica* is a weedy species that opportunistically colonizes disturbed areas. On 17 December 2019, when both MAT and Control tables were already connected to the electric grid, four replicate fragments per species were attached to each tabletop in random order using iron wire. Unfortunately, after about half of the corals were attached to the Control tables, it was realized that the polarity had accidentally been reversed (i.e. the tables were functioning as anode instead of cathode). This error was fixed the same day and outplanting of all other corals was finalized by 19 December. Average start sizes of the fragments ranged between 6–10 cm, depending on the species. Start sizes were equal between treatments for all species, except *P. verrucosa*, for which the initial fragments on the Control tables were significantly larger (see S4 Fig for details). In total, 288 fragments were used in this study (2 treatments x 4 species x 9 replicate tables x 4 replicate fragments per table).

## Measurements

Three indicators were used to track coral performance: the brightness, live tissue cover and growth of fragments. In addition, coral recruits (5–50 mm diameter) were counted on all tables in February 2021 (14 months after deployment). Fragment brightness was used as a proxy for chlorophyll density and therefore (heat) stress resilience [31]. For this, photographs were taken with a Nikon W300 camera on the auto-scene mode. A white slate was held behind a fragment, so that variations in background light intensity could later be corrected. These brightness photographs were taken eight times between March and May 2020 (Fig 1C), covering a marine heatwave that peaked in April. Using the program ImageJ, live parts of coral fragments were selected on these photographs and the average pixel brightness which ranges from 0 (black) to 255 (white) was measured. A representative part of the white slate was also selected and measured. The brightness of a fragment was divided by that of the white slate and multiplied by 255 to standardize for variations in background light intensity. Separate, scaled photographs were taken to quantify the live tissue cover and growth of fragments. These photographs were used to visually quantify the percentage of live coral tissue of each fragment. In addition, these photographs were analysed in ImageJ to determine the maximum length, maximum width and perpendicular width of each fragment, which was then used to determine the fragment's Ecological Volume (EV) as well as the specific growth rate (SGR) for healthy fragments ($\geq$ 80% live coral tissue); See Knoester et al [32] for full details and formulas. The photographs for these live tissue and size measurements were taken seven times throughout the study, starting on 3 February (after the Control treatments were taken off power) and roughly at a bi-monthly interval thereafter (Fig 1C).

## Statistical analysis

All analyses were performed in R [33]. A Beta-regression model with logit link was used to analyse the brightness data using the *glmmTMB* package [34] thereby accounting for the proportional nature of this data, which was scaled between 0 (black) and 1 (white) to allow for model fitting. Treatment, species and date were included as categorical fixed factors and table was included as random factor nested within the power source to account for both the non-independence of the 9 tables connected to the same power source as well as the non-independence of the 4 replicate fragments per species on each table structure. Model assumptions were

validated by visual inspection of *DHARMa* plots [35], the significance of fixed factors evaluated using the *car* package [36] and pairwise mean comparisons were made with Tukey adjustments using *emmeans* [37]. For live tissue cover and growth, the first month (January 2020) was analysed separately, as both the MAT and Control treatment received power this month (S5 and S6 Figs). For consecutive analyses, 22 fragments were excluded (4 on MAT and 18 on the Control) that had a low (< 80%) initial live coral tissue cover, that may be explained by the initial reversed polarity experienced by corals on the Control tables. To iron out any remaining start differences, live coral tissue was standardized to start at 100% by dividing all live coral tissue values across the months by the initial live coral tissue at the start of February (which ranged between 88 to 100%). For the live coral tissue data, the average per species was taken per table to improve model fit. Therefore, a similar Beta-regression model was used as for the brightness data, but without table as random factor. The model assumptions and statistical significance were evaluated as described above. For coral growth, an additional 32 fragments (16 MAT and 16 Control) were omitted from analysis due to clear signs of skeletal predation by fish. A linear mixed-effects model was fit on the square-root transformed growth data using the *lme4* package [38], with treatment, species and date as fixed factors and table as random factor. Adding the power source as hierarchical random factor resulted in a singularity for this model, so the results will be interpreted with caution for pseudo-replication among the nine tables connected to the same power source. Evaluation of residuals and statistical significance were performed as described above. To enable comparisons with prior studies, growth over the full study period is also presented as an increase in total ecological volume (S7 Fig) and linear extension (S8 Fig), using linear mixed-effects models and procedures as described above. Lastly, recruit counts were compared between treatments using a negative binomial generalized linear model from the *MASS* package [39]. Evaluation of the model and statistical significance were performed as described above.

## Results

### Coral bleaching

From March to May 2020, a marine heatwave occurred that accumulated to a temperature anomaly of 10 degree heating weeks in mid-April (S9 Fig) as measured by the NOAA Coral Reef Watch [40]. Bleaching occurred in all four species with a peak in April (Fig 2). All species bleached sooner, longer and more intensely on MAT compared to the Control (Fig 2). However, the effect of treatment over time differed per species, as indicated by a significant three-way interaction ($\chi^2$ = 34.156, *df* = 21, *p* = 0.0349). The difference between treatments was especially clear for *P. cylindrica*, for which coral fragments on MAT were significantly paler on all measured timepoints but one (Fig 2). For *P. verrucosa* and *M. tenera*, the differences were most obvious during the start and peak of the heatwave. For *A. verweyi*, coral fragments on MAT were significantly paler during the peak of the heatwave, but this species did not fully bleach on either treatment (Fig 2). By the end of May, surviving fragments across all four species at least partially regained their colour, yet the difference between MAT and Control treatments was still largely present.

### Live coral tissue

During the peak of the heatwave and its aftermath, live tissue cover dropped across all four species (Fig 3). The effect of treatment over time differed per species, as confirmed by a significant three-way interaction ($\chi^2$ = 92.127, *df* = 18, *p* < 0.001). For *P. verrucosa*, all fragments died (i.e. had 0% live tissue cover) and this happened slightly but significantly earlier on the MAT treatment (Fig 3). Live tissue cover for *P. cylindrica* also strongly declined during the

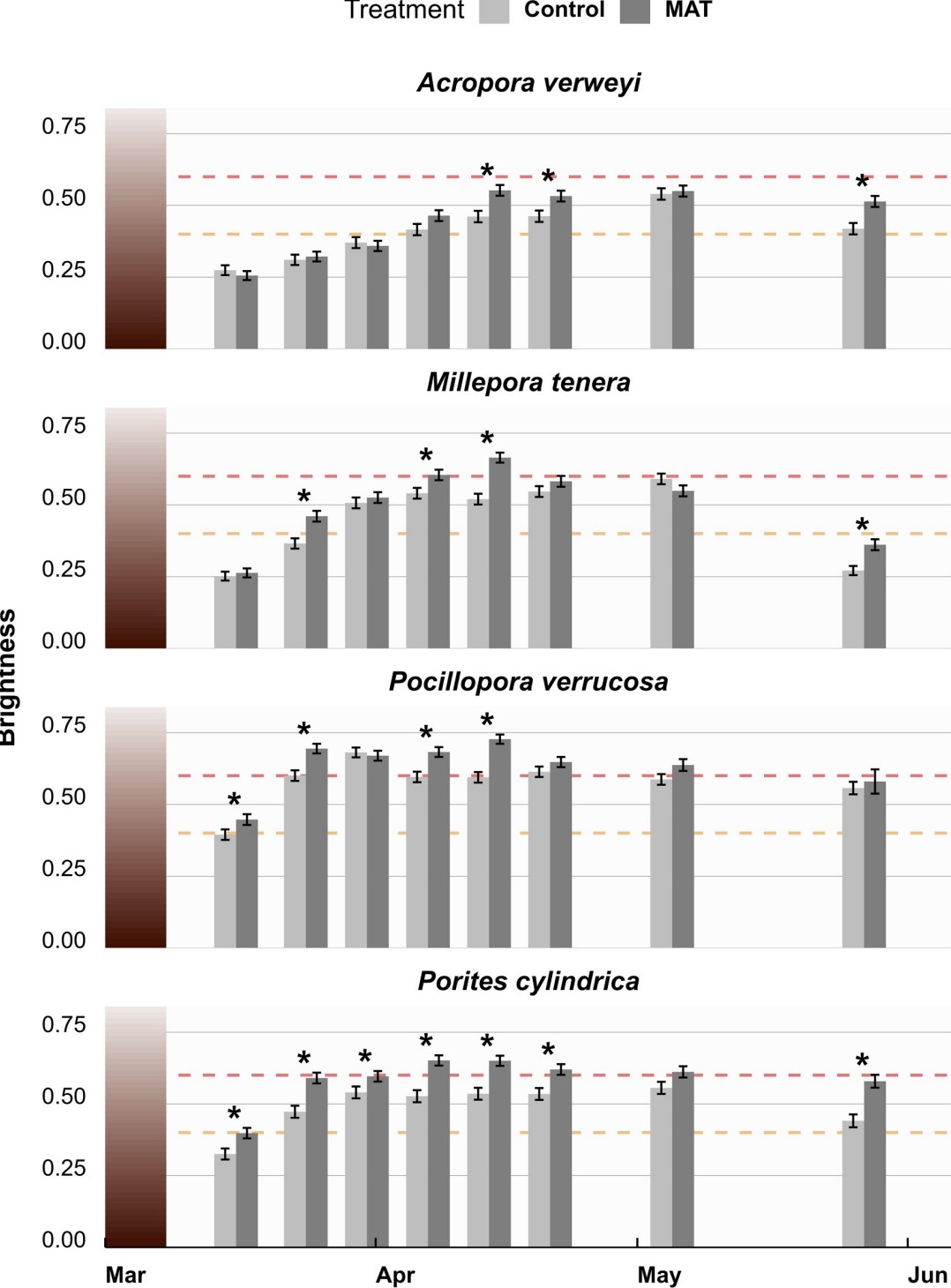

**Fig 2. The brightness of studied coral fragments over time.** Brightness (mean ± SE) is used as indicator for bleaching of the four studied coral species during (March to May 2020) and after (June) a heatwave, compared between the mineral accretion technique (MAT) and Control treatment (n = 9). Brightness is expressed on a scale from 0 (completely black) to 1 (completely white). The dashed orange and red lines indicate the brightness values above which corals are typically said to be paling and bleached, respectively. Significant (p < 0.05) differences in brightness between corals of the MAT and Control treatment are indicated by an asterisk (*) for each time point.

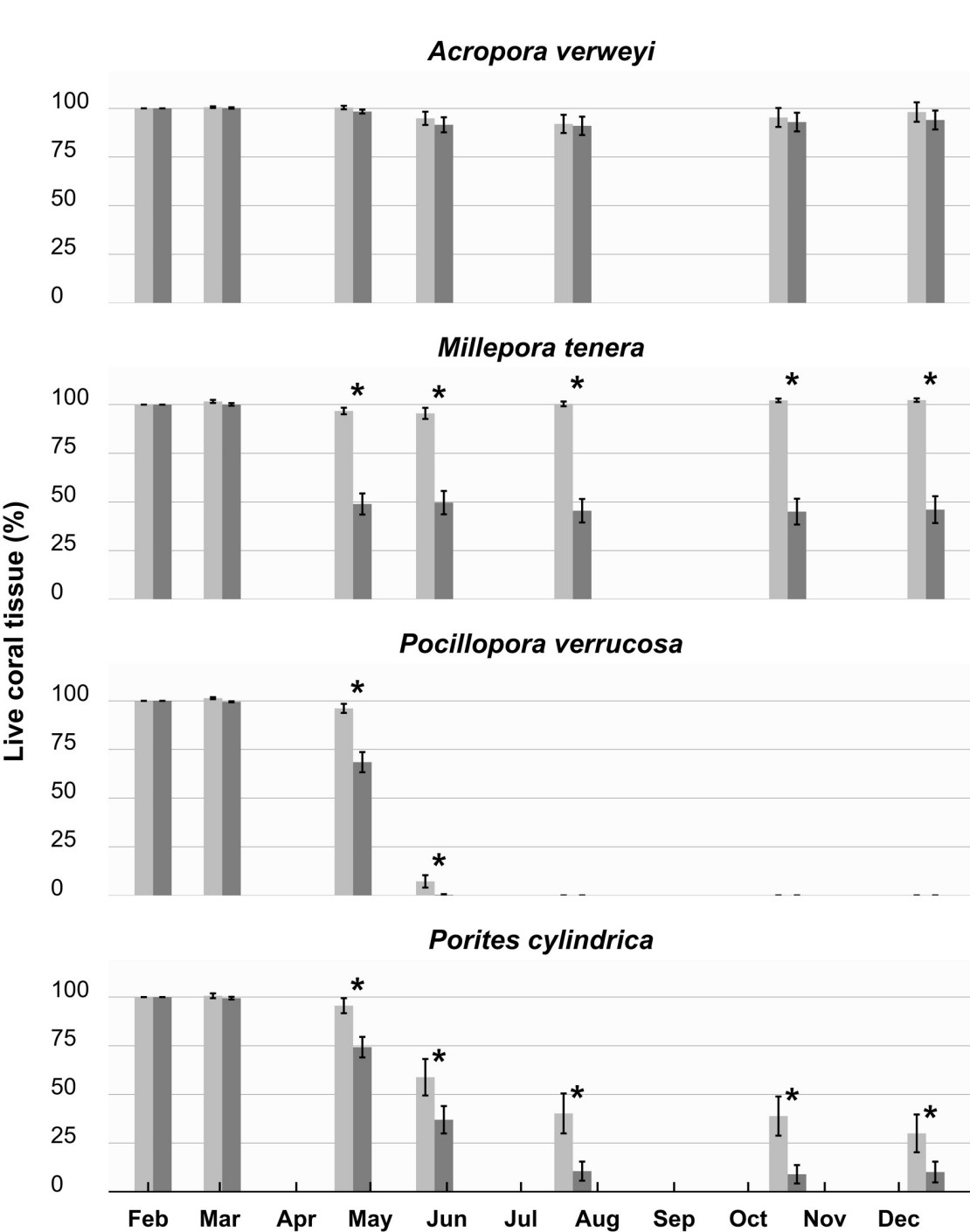

**Fig 3. The condition of studied coral fragments over time.** The percentage (mean ± SE) live coral tissue cover of the four studied coral species throughout the 1-year experiment, compared between the Mineral Accretion Technique (MAT) and Control treatment (n = 9). Significant (p < 0.05) differences in live coral tissue cover between MAT and Control are indicated by an asterisk (*) for each time point. During the study a heatwave occurred between the months March and May.

heatwave, especially on MAT. A significant difference between treatments was sustained up to the end of the experiment, when live coral tissue for *P. cylindrica* was 30 ± 10% (mean ± SE) on the Control and 10 ± 5% on MAT. For *M. tenera*, there was only a negligible decline in live coral tissue on the Control, but a strong decline on MAT. Also this significant difference was sustained to the end of the study, and live coral tissue for *M. tenera* was 102 ± 1% on the Control (i.e. a slight increase compared to the start measurements) and 46 ± 7% on MAT. The pattern for *A. verweyi* was again different from the other species, as only a minor reduction in live tissue cover occurred and no significant difference was found between the treatments. *A. verweyi* ended the experiment with a live tissue cover of 98 ± 5% on the Control and 94 ± 5% on MAT.

## Coral growth

Coral growth rates (Fig 4) dipped during the heatwave and recovered afterwards (except for *P. verrucosa*, as all fragments of this species died). Also for growth, a significant three-way interaction was found between treatment, species and date ($\chi^2 = 33.839$, $df = 11$, $p < 0.001$). After the heatwave, *M. tenera* sustained considerably higher growth rates on the Control compared to MAT (Fig 4). This difference in growth, which was particularly evident directly after the heatwave, was mainly due to quick encrusting growth of *M. tenera* on the Control treatment, which was not seen on MAT (S10 Fig). For the three other species, no consistent differences in growth rates were found between the two treatments over time (Fig 4). Similar patterns between treatments and species were seen when growth was evaluated as total volume produced (S7 Fig) or as linear extension rate (S8 Fig). For both treatments, SGR during the first month of the experiment, when both Control and MAT structures received electrical current, were in the same range as observed in the month after this initial phase, when Control tables stopped receiving electricity (S6 Fig).

## Natural coral recruitment

A large and significant ($\chi^2 = 35.333$, $df = 1$, $p < 0.001$) difference in coral recruitment was observed between Control and MAT tables (Fig 5). A total of 34 recruits were found, and these were exclusively observed on Control tables. These recruits were almost all of the genus *Pocillopora*, with a few *Stylophora* recruits and a single recruit from the Merulinidae family (Fig 5). Not a single recruit was found on any of the MAT tables.

## Discussion

This study examined whether MAT could provide benefits to coral performance before, during and after a marine heatwave. Contrary to both our hypotheses and some earlier studies, no benefits to coral growth, condition or recruitment were found by applying electricity to metal artificial reef structures at commonly-recommended levels of current density. Instead, corals showed either a neutral or negative response to the application of MAT, depending on the species and timing. All four examined species bleached more substantially on MAT, and this resulted in lower live coral tissue cover for three species and a reduced growth rate for one. These negative effects on coral performance on MAT, which became evident only after a marine heatwave at the start of the study, persisted throughout the remaining seven months of the experiment. At the end of the experiment, it had also become clear that coral recruitment was completely impeded on MAT structures. To place these findings in perspective, the remainder of this discussion will focus on two topics: a comparative evaluation of the methodologies used in this study and prior MAT studies, and the remaining benefits that MAT can still provide to reef restoration efforts.

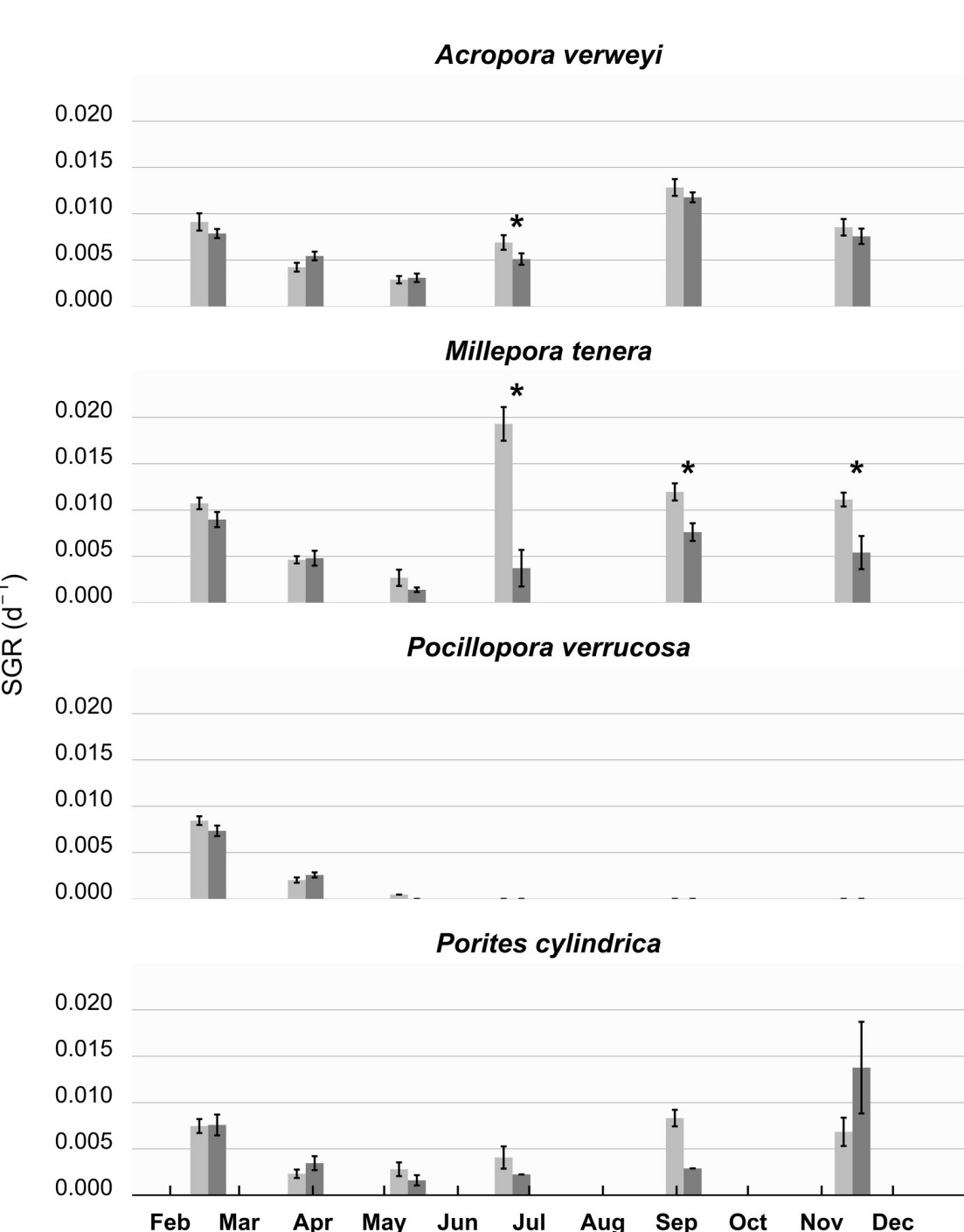

**Fig 4. The growth of studied coral fragments over time.** The mean (± SE) Specific Growth Rate (SGR) of the four studied coral species throughout the 1-year experiment, compared between Mineral Accretion Technique (MAT) and Control treatment (n = 9). Significant ($p < 0.05$) differences in coral growth rates between the MAT and Control treatment are indicated by an asterisk (*) for each time point. During the study a heatwave occurred between the months March and May, which killed all fragments of Pocillopora verrucosa.

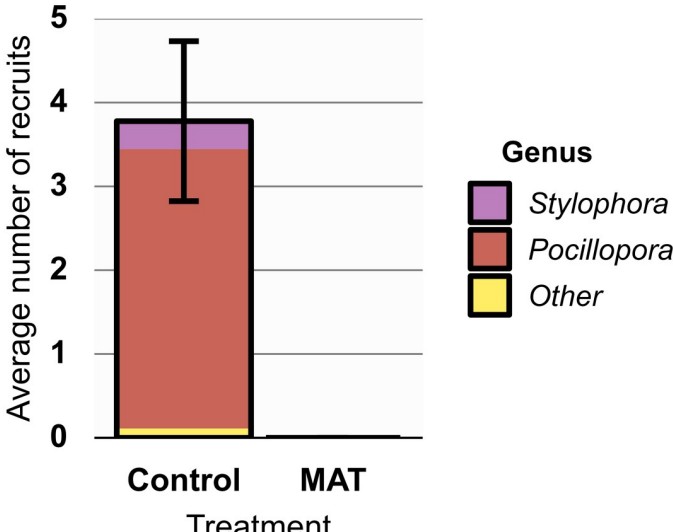

**Fig 5. Hard coral recruits on studied structures.** Mean (± SE) number of hard coral recruits (split per genus) settling on tables of the Control and Mineral Accretion Technique (MAT) treatment (n = 9). Recruits (0.5–5 cm) were counted 14 months after the structures were placed. The surface area of one table was 0.52 m$^2$.

Three methodological considerations can help explain the discrepancy between this study and the reported positive effects of MAT in some earlier studies that applied this method. First and foremost, the better performance of corals on MAT when compared to control structures has often been confounded by additional factors, such as the different design, height or composition of control treatments. For example, several studies compared corals on the elevated MAT structures to control colonies growing on the natural reef [13, 41] or on concrete-based structures [26]. In these studies, the better coral performance on the MAT structures was attributed to the effect of the induced electricity, but could equally well be explained by reduced sedimentation rates, reduced predation by invertebrates, reduced competition or improved water and light conditions due to the raised position of MAT structures above the seafloor. In studies that did use elevated control structures, these were often made of uncoated iron [17, 20, 21, 25, 42–44] and therefore corrosion is likely to have influenced the attachment and possibly also the growth and health of coral fragments on these controls. The few studies that were set up in such a way to prevent these confounding effects, for example by using non-corrosive or pre-coated control structures, have either found negative, mixed, or small and temporary positive effects of MAT on coral performance, the latter especially at low (< 2 A m$^{-2}$) current densities [16, 18, 19, 22].

Therefore, we argue that the second important methodological consideration is the applied current density of MAT. In the original patent, a broad optimal range was given between 0.1 and 30 A m$^{-2}$ [12] and this was later refined to around 3 A m$^{-2}$ through empirical observations by other researchers [45]. More recent experiments indicate that the optimal current density is even lower, with some studies showing overcharging happening already from values as low as 2 A m$^{-2}$ [19, 22, 46]. Others even argue that MAT is only beneficial in the extremely low range of 0.01 to 0.1 A m$^{-2}$ [17, 47]. That could mean that, despite precisely following protocols provided by the founders, practitioners and numerous researchers, the current density used in this study (0.8 A m$^{-2}$) might have also been too strong, potentially explaining the observed negative effects on coral performance. Regrettably, a large number of studies do not report the

current density used [e.g. 20, 25, 26, 43, 44, 48], making effective comparisons nearly impossible. Clearly, identifying and reporting the current density at which MAT could provide benefits to coral performance should be a research priority and will likely help to resolve conflicting findings of earlier studies. As the current density is expected to decrease over time due to the isolating effect of the accretion layer [23, 49], monitoring this factor over time is also important.

A third explanation for the inconsistent results between studies could be due to any species and site specific differences. Previous studies have already shown that even closely-related species can differ in their response to MAT [18]. Such species-specific differences were confirmed in this study: though all examined species showed negative responses to MAT during a heatwave, the severity of their responses varied significantly. Interestingly, the most negative response was found for the hydrozoan *Millepora*—the species that experienced the most direct contact with the MAT structures due to its encrusting growth form. In prior studies there has been little overlap in species used, however, making it difficult to distinguish if differences between studies are due to species or site specific effects. Even though it has been claimed that MAT works under a wide range of conditions [42], quantification is lacking and it can be reasonably expected that differences in for example alkalinity, salinity, temperature or nutrients can influence the effects of MAT. While complete standardization across studies is impractical, at least some overlap in species and conditions will help to disentangle the relative importance of these factors. It goes without saying that study reports should at the very least include information on the local environmental conditions. Lastly, operating independently replicated MAT setups (each with their own power cable and supply) would allow a more broad evaluation of the method. Both the current study and many of the prior studies were confined to a single location using a single power source and could therefore be easily influenced by small-scale differences in local factors such as the fish community (see also notes on coral predation by fish in Goreau, 2014 [50]).

The variation in experimental design, species, conditions and reporting completeness of prior studies have all contributed to the remaining uncertainty on the effectiveness of MAT. Nonetheless, several patterns emerge and are supported by this study. First, any potential benefits to coral growth are likely to be dependent on the applied current density, with negative effects becoming increasingly likely at higher current densities. Properly replicated and controlled experiments are needed at various current densities to determine if small improvements in coral growth are indeed possible, and if so, to elucidate the underlying physiological mechanisms within the coral or its microbiome. Second, the direct effect of electricity is unlikely to provide any health benefits or heat resilience to corals under the commonly recommended settings for MAT. Instead, corals appear to have typically fared well on MAT structures due to the solid attachment and elevated position. Third, constant electricity has a clear negative effect on coral recruitment, potentially by delaying biofilm development, engulfing new recruits or by repelling settling planula [23, 46, 51]. Thus, unfortunately, MAT does not represent a panacea to safeguard coral reefs by significantly enhancing coral growth, heat resilience or recruitment when applied under the commonly recommended settings.

Nonetheless, MAT can still provide tangible benefits to coral reef restoration projects for at least three reasons. First, the accreted calcium-carbonate layer is an effective measure to reinforce the attachment of outplanted coral fragments. Second, the accreted layer (without continued electricity) is a suitable substrate for coral recruitment. Third, a high diversity of artificial reef structures can be easily created against low carbon emissions and without the need for polluting anti-corrosion coatings. Indeed, these benefits in combination with the typically elevated design of MAT structures could explain observed successes with this technique, which are now inadvertently attributed to large physiological benefits to coral

performance through the electricity itself [13]. As it appears that the main, indirect, benefits of electricity manifest themselves in the initial months (i.e. providing coral attachment and an anti-corrosion layer) and as coral recruitment is hindered by the effects of continuous electricity, a more effective use of MAT installations could be to power structures only during the first weeks after installation and then move them off the electric grid. This would enable more cost-effective use of the proven benefits of MAT and facilitate upscaling of reef restoration efforts. Such a short-term application of MAT could improve reef restoration techniques and, when combined with local stress reduction and climate action, contribute to the conservation and rebuilding of functional coral reefs [52].

## Supporting information

**S1 Checklist. Inclusivity checklist.**
(DOCX)

**S1 Fig. Map of the study area for the Mineral Accretion Technique (MAT) experiment.** Indicated are the position of the MAT (-4.650239, 39.387031) and Control table patches (-4.649854, 39.387534), relative to the power source and the nearby village Shimoni. Imagery from Google Earth (Image © 2024 Airbus).
(DOCX)

**S2 Fig. Pictures of the experimental setup.** A: A MAT (Mineral Accretion Technique) table with power cables visible running underneath. B: Top view of a Control table with the 16 fragments attached. C: Close-up of the power cable splitting into 9 smaller cables running to each of the MAT tables. D: Close-up of the mid-water suspended titanium anode. E: Close-up of a Pocillopora fragment attached to a MAT table a few days ago, showing initial deposition of calcium carbonate. F: Close-up of table structure after 1 year showing the thick calcium carbonate layer deposited.
(DOCX)

**S3 Fig. Mean (± SE) circumference of the metal rebar stakes that made up the tables of the Control and Mineral Accretion Technique (MAT) treatments after 14 months (n = 9).** Both the Control and MAT tables were charged for the first month of the experiment, after which the Control tables were taken off the grid and moved 35 m away. The dashed line shows the initial circumference at the start of the study (for both Control and MAT tables). Converted to diameters (i.e. thickness), the initial diameter was 1.4 cm and, after 14 months, the diameter was 1.6 cm for the Control tables and 2.9 cm for MAT tables.
(DOCX)

**S4 Fig. The average start sizes expressed as Ecological Volume (EV) of the four studied coral species, compared between Mineral Accretion Technique (MAT) and Control tables.** Error bars show SE (n = 9). A significant difference in start size was found between treatments using a linear model ($\chi 2$ = 7.992, df = 1, p = 0.00470). A Tukey post hoc test revealed that the start volume of Pocillopora verrucosa was significantly higher in the Control (p < 0.05, indicated by asterisk), whereas no significant differences were found for the other three species.
(DOCX)

**S5 Fig. The percentage (mean ± SE) of live coral tissue of the four studied coral species after the first month (January 2020), compared between Mineral Accretion Technique (MAT) and Control treatment (n = 9).** During this first month, the Control tables received the same amount of electricity as the MAT tables to form an anti-corrosion layer. No significant differences in live coral tissue were found between treatments for any species in this first

month ($X^2$ = 7.05, df = 3, p = 0.0702).
(DOCX)

**S6 Fig. The mean (± SE) Specific Growth Rate (SGR) of the four studied coral species during the first month of the study (December 2019 –January 2020), compared between Mineral Accretion Technique (MAT) and Control structures (n = 9).** During this first month, the Control tables received the same amount of electricity as the MAT tables to form an anti-corrosion layer. No significant differences in growth rates were found between the treatments for any species in this first month ($X^2$ = 1.86, df = 3, p = 0.601).
(DOCX)

**S7 Fig. The mean (± SE) percentage increase in Ecological Volume (EV) of the four different studied coral species (*Acropora verweyi, Millepora tenera, Pocillopora verrucosa and Porites cylindrica*) at the end of the 1-year experiment, compared between Control and Mineral Accretion Technique (MAT) treatment (n = 9).** For example, an EV increase of 200% means the fragment volume doubled, an EV increase of 100% means the fragment remained the same volume and an EV increase of 0% means a fragment lost all its volume (i.e. died). Note the logarithmic scale. A heatwave during the experiment killed all *P. verrucosa* fragments. Significant (p < 0.05) differences in EV increase between the MAT and Control treatment are indicated by an asterisk (*) for each species.
(DOCX)

**S8 Fig. The mean (± SE) Linear Extension Rates (LER in cm y-1) throughout the study period for the four studied coral species (*Acropora verweyi, Millepora tenera, Pocillopora verrucosa and Porites cylindrica*), compared between the Control and Mineral Accretion Technique (MAT) treatment (n = 9).** Decreases in live coral tissue resulted in negative LER for some species. Significant differences in EV increase between the MAT and Control treatment are indicated by asterisks (*p < 0.05; ** p < 0.01; *** p < 0.001) for each species.
(DOCX)

**S9 Fig. Overview of water temperature and heat stress during the one-year study.** The light blue line shows sea surface temperature as determined by NOAA from satellite data. The dark blue line shows the measured water temperature at the study location from March to September (using HOBO loggers). The dashed yellow line shows the long-term yearly mean maximum monthly temperature (MMM) plus 1 degree: this is used by NOAA as heat stress threshold. The dashed red line shows the accumulated heat stress as Degree Heating Weeks (DHW), following NOAA's Coral Reef Watch (Liu et al., 2006).
(DOCX)

**S10 Fig. Representative photographs showing the differences in growth of the hydrozoan *Millepora tenera* on the Mineral Accretion Technique (MAT) and Control tables.** These pictures show growth in the month (June–July 2020) directly following a heatwave, when differences between MAT and Control were largest. Note the quick growth and encrusting of the coral on the Control table. Also note that the iron wire to attach the coral is overgrown on the Control table, but tissue around this wire died on MAT.
(DOCX)

## Acknowledgments

We would like to thank Pilli Pipa Dhow Safaris for their logistic support and for providing the electricity for the experiment. We thank Francesca Sangiorgi for assisting with supervision

and Kyra Pikaar for creating S9 Fig. A special thanks to Mgeni Wamwachai, Yatin Patel, Hamadi Mwamlavya, Bart Schoon, Susanne Bähr, Anne Wolma, Rosanne Bartholomeus, Renee Melkert, Luis Almeida, Omar Salim, Quirijn Schürmann, Tim Lardinois and Sasha Koning for assisting with the fieldwork.

## Author Contributions

**Conceptualization:** Ewout Geerten Knoester, Albertinka J. Murk, Ronald Osinga.

**Data curation:** Richard Sanders, Daisy Durden, Bulisa O. Masiga.

**Formal analysis:** Ewout Geerten Knoester.

**Investigation:** Ewout Geerten Knoester, Richard Sanders, Daisy Durden, Bulisa O. Masiga.

**Methodology:** Ewout Geerten Knoester, Albertinka J. Murk, Ronald Osinga.

**Project administration:** Ronald Osinga.

**Resources:** Albertinka J. Murk.

**Supervision:** Albertinka J. Murk, Ronald Osinga.

**Visualization:** Ewout Geerten Knoester.

**Writing – original draft:** Ewout Geerten Knoester.

**Writing – review & editing:** Richard Sanders, Daisy Durden, Bulisa O. Masiga, Albertinka J. Murk, Ronald Osinga.

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
