## [Decision Letter · Decision Letter 0]

25 Sep 2024

PONE-D-24-32565Lack of positive effects by mineral accretion technique on the bleaching resilience, growth and recruitment of coralsPLOS ONE

Dear Dr. Knoester,

Thank you for submitting your manuscript to PLOS ONE. After careful consideration, we feel that it has merit but does not fully meet PLOS ONE’s publication criteria as it currently stands. Therefore, we invite you to submit a revised version of the manuscript that addresses the points raised during the review process.

We look forward to receiving your revised manuscript.

Kind regards,

Satheesh Sathianeson, Ph.D

Academic Editor

PLOS ONE

**Journal Requirements:**

Reviewers' comments:

Reviewer's Responses to Questions

**Comments to the Author**

1. Is the manuscript technically sound, and do the data support the conclusions?

Reviewer #1: Yes

Reviewer #2: Yes

2. Has the statistical analysis been performed appropriately and rigorously? 

Reviewer #1: No

Reviewer #2: Yes

3. Have the authors made all data underlying the findings in their manuscript fully available?

Reviewer #1: Yes

Reviewer #2: Yes

4. Is the manuscript presented in an intelligible fashion and written in standard English?

Reviewer #1: Yes

Reviewer #2: Yes

5. Review Comments to the Author

**Reviewer #1:** Dear authors, dear editor,

The study “Lack of positive effects by mineral accretion technique on the bleaching resilience, growth and recruitment of corals” by Knoester and co-authors has investigated the effect of the Mineral Accretion Technique (MAT) on bleaching, live tissue cover, growth and recruitment of four coral species in Kenya over one year. They found more bleaching and lower growth rates of corals on MAT structures compared to control structures.

This manuscript is very well written and describes the study and results very clearly (including nice visualisations of the results). The introduction includes all necessary background information about the MAT method. The discussion provides a very nice and concise comparison of the results with previous studies. My only major comment about this study is about the study design where the nine tables of the MAT setup were not true replicates as they have all been connected to the same power supply. This should be clarified in the text and also taken into account for the statistical analysis of the data. Apart from this, I only have some minor comments, especially in the methods sections where the authors should clarify a few things.

The authors have provided all their data on GitHub, however I would recommend submitting the data to a data repository such as PANGAEA where the data files are checked according to the FAIR principle instead.

Abstract

The abstract provides a good summary of the study. However, due to the findings of the study, I would recommend re-rephrasing the title to “Negative effects of mineral accretion technique (MAT) on the bleaching resilience, growth and recruitment of three coral species” as the authors state that they didn’t just find no effect of MAT but even a significantly negative effect compared to the control.

Introduction

The introduction is very well written and provides all necessary background information and controversial previous findings about this method for reef rehabilitation. It clearly states why this study is needed and the study aims as well as hypotheses.

Line 38: I would suggest to include some more information about the reasons why coral reefs have declined globally.

Line 81: Change references to numbers for consistency.

Line 86: What do you mean by “condition”? Please clarify.

Lines 86-89: Please include information where this study took place.

Methods

I think a map of the study site with the location of the MAT structures and controls would be helpful.

Line 100: I find it quite interesting that the pH varies so much at this site, can you give an explanation for this?

Lines 111-112: In my opinion, the nine tables of the MAT treatment cannot be viewed as true replicates if they were all connected to the same electricity source as indicated in Figure 1. Why was there not an individual electricity supply used for each table? Please clarify this in the text and also take it into account for the statistical analysis of the data.

Line 114: Please re-phrase as “was moved outside the electrical field of the MAT structure for the remainder of the study” to make it more clear, if I understood this correctly.

Lines 120-121: Unclear why the control tables were replaced and additional metal structures were connected to the grid, please specify.

Figure 1: Unclear what “5 Pod MAT” is, please clarify. In the timeline (C), it looks like coral brightness was only investigated during the marine heatwave, why was this not continued during the rest of the experiment to investigate coral recovery after the heatwave?

The schematic of the experiment setup is very nice and helpful, however I think it would also be helpful for the readers to see actual photos of the structures/setup under water (or on land if underwater pictures are not available). Please provide some photos of the setup in the supplements if possible.

Line 129: Unclear what the difference between the tables and pods are, please clarify.

Line 142: Please define “short power outages”. How often and for how long did they occur?

Line 143-145: Are there any information about how MAT affects other organisms than corals as it might also be beneficial for e.g. growth of coralline algae, which might compete with corals for space?

Line 161: The nine tables cannot be treated as true replicates if they were all connected to the same power supply, please see comment above.

Line 170: Why was the investigation of the brightness of coral fragments not continued after the heatwave/during the whole duration of the study?

Line 178-181: How was the growth rate of the corals determined? I guess also by using the pictures and this is meant by “size measurements”? Please give some more details about the growth measurements.

Line 183: I would suggest to change it to “statistical analysis”.

Lines 187-189: Please also take into account that the nine tables cannot be treated as true replicates as they were all connected to the same power supply, therefore not only the 4 fragments on each table but also the 9 tables of each treatment are not independent. This also needs to be taken into account for the analysis of all other parameters. In addition, average values for each table should be used for the statistical analysis of all parameters and with table as random factor, not only for live tissue data to account for the independence of the 4 coral fragments on each table for all parameters.

Lines 204-205 and 208-211: I would move the description of growth measurements to the previous paragraph. Convert reference to number for consistency.

Line 210: I would suggest to also plot the linear extension rate data instead of presenting the data in a table.

Results

Lines 219-220: I would suggest to write “bleaching occurred in all four species with a peak in April” instead.

Figure 3: I would suggest to clarify the figure legend by not calling it “condition” but “live coral tissue cover”.

Discussion

Lines 315-316 and 340: Change references to numbers for consistency.

Line 314: I would suggest to delete “shockingly”.

**Reviewer #2: **Largely well written and a well-planned experiment. A few minor types and grammatical issues plus some more detail should be added to methods and a few considerations for discussion that I think are warranted prior to publication.

L100: remove ppt; Salinity is unitless. Also remove "between" before ranges with the en dashes throughout manuscript: salinity was 34-35m PH was 7.8-82, conductivity was...

L103: use abbreviation for meters after roman numerals

L118: delete "between, and use date format of 17-19 December 2019; fix throughout manuscript

L129: 235 V

L148: using a comma before the last item in a series throughout would clarify complex series, such as this one. I would reorder to put the series in the series last for clarity.

L172, which clause out of place; move to after brightness

L195: the which clause should be a that clause without a comma: ..live coral tissue cover that may be explained by... (delete mainly)

L183-214: are there options you need to provide for any of these packages? If so, please provide the settings you used or make your code available so the analyses are repeatable by someone else

L218-219: is something missing here: 10 degree heating weeks?

L227: this needs an antecedent; so they did not fully bleach on MAT? or other species did not or did frequently bleach? Please clarify this section

L235: however in the middle of sentences is not necessary and are a bit confusing as they seem to present contrary information but you are using them even for supporting information. I noted this issue elsewhere but did not comment. Please review your use of however throughout.

L276: not a single benefit to coral performance that was evaluated in this study was found.... (unless you examined all possible benefits you should specify here so it is less likely you will be misquoted)

Results issues that need to be addressed and some thought for consideration. Given the timing of the warming/bleaching event, I think you need to have a few more caveats in there about the possibility of that event confounding the results for live coral tissue and growth rate. None of the differences between control and MAT were significant before the bleaching event. Therefore, the large differences you observed may not have been present if not for the apparent negative affect of MAT on bleaching, which may not be an effect MAT has on the coral but their zooxanthellae or other symbionts. I think other studies should look at whether the microbiome differs on control vs MAT. Does current have negative effects on single cells but lesser on multicellular organisms and thus the colonies?

6. PLOS authors have the option to publish the peer review history of their article (what does this mean?). If published, this will include your full peer review and any attached files.

Reviewer #1: No

Reviewer #2: **Yes**

---

## [Author Response · Author response to Decision Letter 0]

8 Nov 2024

We thank the reviewers for their time and effort taken to give constructive feedback to our manuscript. Please see the document 'Response to Reviewers' for our detailed responses to the received feedback.

---

## [Decision Letter · Decision Letter 1]

27 Nov 2024

Negative effects by mineral accretion technique on the heat resilience, growth and recruitment of corals

PONE-D-24-32565R1

Dear Dr. Knoester,

We’re pleased to inform you that your manuscript has been judged scientifically suitable for publication and will be formally accepted for publication once it meets all outstanding technical requirements.

Kind regards,

Satheesh Sathianeson, Ph.D

Academic Editor

PLOS ONE

Additional Editor Comments (optional):

Reviewers' comments:

Reviewer's Responses to Questions

**Comments to the Author**

1. If the authors have adequately addressed your comments raised in a previous round of review and you feel that this manuscript is now acceptable for publication, you may indicate that here to bypass the “Comments to the Author” section, enter your conflict of interest statement in the “Confidential to Editor” section, and submit your "Accept" recommendation.

Reviewer #1: All comments have been addressed

2. Is the manuscript technically sound, and do the data support the conclusions?

Reviewer #1: Yes

3. Has the statistical analysis been performed appropriately and rigorously? 

Reviewer #1: Yes

4. Have the authors made all data underlying the findings in their manuscript fully available?

Reviewer #1: Yes

5. Is the manuscript presented in an intelligible fashion and written in standard English?

Reviewer #1: Yes

6. Review Comments to the Author

Reviewer #1: (No Response)

7. PLOS authors have the option to publish the peer review history of their article (what does this mean?). If published, this will include your full peer review and any attached files.

Reviewer #1: No

---

## [Editor Report · Acceptance letter]

6 Dec 2024

PONE-D-24-32565R1 

PLOS ONE

Dear Dr. Knoester, 

I'm pleased to inform you that your manuscript has been deemed suitable for publication in PLOS ONE. Congratulations! Your manuscript is now being handed over to our production team.

Kind regards, 

on behalf of

Dr. Satheesh Sathianeson 

Academic Editor

PLOS ONE